# Identification and Characterization of WOX Gene Family in Flax (*Linum usitatissimum* L.) and Its Role Under Abiotic Stress

**DOI:** 10.3390/ijms26083571

**Published:** 2025-04-10

**Authors:** Xixia Song, Jianyu Lu, Hang Wang, Lili Tang, Shuyao Li, Zhenyuan Zang, Guangwen Wu, Jian Zhang

**Affiliations:** 1Heilongjiang Academy of Agricultural Sciences, Harbing 150086, China; jjzwyjs@163.com (X.S.); tanglili19861126@126.com (L.T.); 2Faculty of Agronomy, Jilin Agricultural University, Changchun 130118, China; lu@mails.jlau.edu.cn (J.L.); 20240795@mails.jlau.edu.cn (H.W.); 13943374270@163.com (S.L.); zhenyuanzang1989@163.com (Z.Z.); 3Department of Biology, University of British Columbia, Kelowna, BC V1CIV7, Canada

**Keywords:** flax, WOX, gene family, phylogeny analysis, abiotic stress

## Abstract

The WOX (WUSCHEL-related homeobox) gene family plays pivotal roles in plant growth, development, and responses to biotic/abiotic stresses. Flax (*Linum usitatissimum* L.), a globally important oilseed and fiber crop, lacks a comprehensive characterization of its WOX family. Here, 18 *LuWOX* genes were systematically identified in the flax genome through bioinformatics analyses. Phylogenetic classification grouped these genes into three clades: Ancient, Intermediate, and WUS Clades, with members within the same clade exhibiting conserved exon–intron structures and motif compositions. Promoter analysis revealed abundant cis-acting elements associated with hormone responses (MeJA, abscisic acid) and abiotic stress adaptation (anaerobic induction, drought, low temperature). Segmental duplication events (nine gene pairs) contributed significantly to *LuWOX* family expansion. Protein–protein interaction networks implicated several *LuWOX* proteins in stress-responsive pathways. Expression profiling demonstrated that most *LuWOX* genes were highly expressed in 5-day-post-anthesis (DPA) flowers and embryonic tissues. qRT-PCR validation further uncovered distinct expression patterns of *LuWOX* genes under cold, drought, and salt stresses. This study established a foundational framework for leveraging *LuWOX* genes to enhance stress tolerance in flax breeding and functional genomics.

## 1. Introduction

The WUSCHEL-related homeobox (WOX) gene family is a plant-specific transcription factor family. Its conserved sequence consists of a homeodomain (HD) formed by approximately 60–66 amino acids arranged in a “helix-loop-helix–turn–helix” structure. WOX transcription factors (TF) bind to specific DNA sequences via their HD to exert biological functions, and they are extensively involved in plant growth and development, the regulation of stem cell division and differentiation, and processes such as embryo and organ formation [1]. Through the phylogenetic analysis of WOX proteins across diverse plant species, the WOX gene family is classified into three clades: the WUS Clade, Intermediate Clade, and Ancient Clade [2].

The WUS gene was first discovered in *Arabidopsis*, which is mainly expressed in the meristematic tissue of the stem tip and plays an important role in the formation and maintenance of SAM [3]. Genetic and molecular biology studies on WOX genes have shown that WOX members play important roles in various physiological and developmental processes of plants [4,5]. Some studies have found that the WUS gene of maize and the regeneration gene BBM play a key role in the process of plant tissue culture, and the co-transformation of the two genes can significantly improve the transformation efficiency of maize, sorghum, and sugarcane [6]. The *WOX1* and *WOX3* genes play crucial roles in regulating lateral organ growth across diverse plant species including *Arabidopsis*, *Cucumis sativus*, and *Panicum virgatum* [7,8,9]. The *WOX2* transcription factor performs conserved regulatory functions in precursor cell specification during zygotic embryogenesis, with phylogenetic analysis revealing that its ortholog *PaWOX2* in *Picea abies* retains functional equivalence as a molecular marker for monitoring embryogenic competence acquisition in gymnosperm systems [10]. *WOX4* can promote the differentiation of vascular cambium in *Arabidopsis* and tomato [11]. *WOX5* is expressed in the root apical meristem (RAM) and maintains the stability of stem cells in RAM [12]. In *Arabidopsis*, the transcription factor *AtWOX7* exhibits constitutive expression during all developmental phases of lateral root organogenesis while functionally demonstrating negative regulation of lateral root primordium formation through carbohydrate-availability-dependent modulation. Furthermore, the overexpression of *AtWOX7* significantly reduces the number of lateral root primordia [13]. The *WOX11* and *WOX12* genes exhibit functional redundancy in regulating root system development. The overexpression of *PagWOX11/12a* significantly promotes root elongation and enhances biomass accumulation in *Populus alba × glandulosa* [14]. The *WOX13* and *WOX14* genes are associated with floral organ formation and development. The overexpression of *OsWOX13* in rice advances the flowering time by 7–10 days [15]. The *AtWOX14* transcriptional regulator mediates the developmental trajectory of vascular precursor cells in *Arabidopsis* inflorescence stems, acting through the coordinated activation of protoxylem/metaxylem specification programs and phenylpropanoid biosynthesis networks to drive cellular lignification during secondary cell wall deposition [16]. The evolutionarily conserved *PFS2/WOX6* transcriptional regulator in *Arabidopsis* exhibits spatiotemporal expression specificity synchronized with megasporogenesis and integument morphogenesis, mechanistically orchestrating gametophytic phase transition through the precise modulation of ovule patterning networks [17,18]. In cucumber, the *CsWOX9* gene is predominantly expressed in developing fruits, and its overexpression in *Arabidopsis* resulted in a shortened stem phenotype in a study [19].

The WOX gene family is implicated in dual roles: regulating plant growth and development and mediating stress response mechanisms. Specifically, *AtWOX6* plays a critical role in root apical meristem maintenance and root development while also mediating cold stress responses [20]. The *GmWOX* gene in soybean responds to different abiotic stress treatments (cold and drought) [21]. The overexpression of the rice *OsWOX13* gene can improve drought tolerance in rice [15]. Knocking out *GhWOX4* in cotton significantly reduces drought tolerance in transgenic cotton [22]. The overexpression of *PagWOX11/12a* gene significantly enhances drought and salt tolerance in poplar trees [23]. WOX genes play roles in biotic stress; *WOX11* restricts cyst nematode parasitism by modulating ROS-mediated cell wall plasticity in syncytia, independent of its known role in adventitious root-mediated tolerance. This dual functionality highlights *WOX11* as a key regulator balancing plant survival and pathogen suppression [24].

Flax, a dual-purpose crop domesticated from paleobotanical origins for phloem-derived textile fibers and lipid-rich seeds, maintains contemporary agronomic significance with extensive cultivation spanning temperate latitudes to support global linseed oil production and bast fiber industries [25]. According to their ultimate use, flax plants are divided into three categories: oils, fibers, and dual-use [26]. Flaxseed is rich in essential nutrients, including lignans, dietary fiber, and high-quality protein, with alpha-linolenic acid (ALA), an omega-3 fatty acid, serving as a key component of human dietary requirements [27]. To date, no systematic studies on the WOX gene family in flax have been reported. This study employed bioinformatics approaches to comprehensively investigate flax WOX genes, including their evolutionary relationships, genomic distribution, physicochemical properties, sequence features, phylogeny, promoter cis-regulatory elements, protein interaction networks, and expression patterns. Key WOX genes were identified as responsive to low-temperature and drought stress conditions, providing a foundation for the further exploration of their biological functions in flax. This paper represents the first comprehensive report on WOX genes in flax, offering critical insights into their roles in development and abiotic stress adaptation.

## 2. Results

### 2.1. Phylogenetic Characterization and Identification of WOX Family Genes in Flax

A total of 18 WOX genes (*LuWOX1*–*LuWOX18*) were identified in the flax genome (longya10) through BLASTP alignment and PF00046 Hidden Markov Model (HMM) analysis (Table 1). The characterization of their encoded proteins revealed significant physicochemical diversity: *LuWOX14* exhibited the longest sequence (432 amino acids) whereas *LuWOX15* displayed the shortest (192 amino acids), with molecular weights ranging from 21.81 to 47.91 kDa. Notably, 66.67% of *LuWOX* proteins displayed a pI > 7, indicating a predominance of basic amino acids. Stability predictions classified *LuWOX01* as the only stable protein (Instability Index: 39.55–74.39) while Aliphatic Index values (47.40–70.10) and Grand Average of Hydropathicity (GRAVY) analysis confirmed that all *LuWOX* proteins were hydrophobic. Subcellular localization unanimously predicted nuclear targeting, consistent with these proteins’ putative roles as transcription factors. These findings collectively provide a foundational framework for further functional studies on *LuWOX* genes in flax development and stress adaptation.

To elucidate the phylogenetic topology of the WOX gene family in flax, a maximum likelihood tree was inferred from aligned amino acid sequences of *Arabidopsis* (16), *Oryza* (14), and *Linum* (18), comprising 48 conserved WOX orthologs (Figure 1) (Appendix A). Based on the established *Arabidopsis* WOX subfamily classification, the 18 *LuWOX* genes were phylogenetically categorized into three clades: the Ancient Clade, Intermediate Clade, and WUS Clade. The WUS Clade contained the majority of members (12 genes: *LuWOX7*–*LuWOX16*) while the Intermediate and Ancient Clades comprised fewer genes. Comparative analysis revealed closer evolutionary relationships between flax and *Arabidopsis* WOX families than those with rice, suggesting lineage-specific conservation. This phylogenomic framework advances our understanding of WOX gene evolution in flax and provides a basis for cross-species functional studies.

### 2.2. Analysis of Gene Structure and Conservative Motif of LuWOX

To delineate structural conservation within the flax WOX family, 10 conserved motifs (motif1–motif10) were identified in 18 *LuWOX* proteins using MEME suite analysis (Figure 2A,B). Phylogenetic reconstruction based on motif alignment mirrored the clade classification in Figure 1, confirming evolutionary coherence. Motif composition analysis revealed that motif1, representing the core WOX domain, was universally present across all *LuWOX* proteins. Notably, motif8 was detected in all members except *LuWOX7* while motif2 was absent solely in *LuWOX16*. Subfamily-specific motif patterns were evident: within the WUS Clade, motif1, motif2, motif3, and motif8 co-occurred in all members except *LuWOX7* and *LuWOX16*, which lacked specific motifs (Appendix A). Motif lengths ranged from comprising nine to fifty amino acids, with positional variability (sites 2–18) reflecting functional diversification. These structural insights underscore the evolutionary conservation and subfunctionalization of *LuWOX* genes, providing a basis for probing their regulatory roles in flax development. The exon–intron structure analysis of *LuWOX* genes revealed that exon numbers ranged from two to four while intron numbers varied between one and three (Figure 2C). Six genes (*LuWOX5*, *LuWOX6*, *LuWOX13*, *LuWOX14*, *LuWOX17*, and *LuWOX18*) exhibited the highest structural complexity, containing four exons and three introns. Notably, genes within the same clade shared highly similar exon numbers, suggesting functional conservation across phylogenetically related members.

### 2.3. Chromosome Localization and Gene Reproduction Events of LuWOX Gene Family

The chromosomal localization analysis of the 18 *LuWOX* genes within the flax reference genome revealed their uneven distribution across 12 chromosomes (Figure 3). Chromosomes 5 and 11 harbored the highest number of genes (three each, 16.67% of the total), followed by chromosomes 2 and 8 (two genes each, 11.11%). The remaining chromosomes (1, 6, 7, 9, 10, 12, 13, and 15) each contained only one *LuWOX* gene (5.56% per chromosome). This distribution pattern highlights potential hotspots for gene duplication or evolutionary divergence in flax.

The gene duplication analysis of the *LuWOX* family using BLAST 2.14.0 and MCScanX revealed no tandem duplication events but identified nine segmentally duplicated *LuWOX* gene pairs (Figure 4A), suggesting that whole-genome duplication likely drove the expansion of this family. We also found that all collinear gene pairs of *LuWOX* had Ka/Ks ratios of less than 1, indicating that they had evolved under purifying selection (Table 2). Comparative synteny analysis with *Arabidopsis*, maize, and rice further uncovered ten, ten, and six collinear gene pairs, respectively (Figure 4B). Specifically, *Arabidopsis* chromosomes 2, 3, and 5 exhibited collinearity with flax chromosomes 1, 2, 7, 8, 9, 12, and 15; maize chromosomes 3, 6, and 8 aligned with flax chromosomes 2, 5, 8, 11, and 15; and rice WOX genes showed synteny with flax chromosomes—2, 5, 8, 11, and 15. While conserved collinear regions were observed across species, lineage-specific variations in gene pair distribution underscored the dynamic evolutionary trajectory of the *LuWOX* family. These findings highlight the role of WGD and divergent selection in shaping the structural and functional diversification of WOX genes in flax.

### 2.4. Analysis of Cis-Acting Elements

To elucidate the regulatory mechanisms of *LuWOX* genes under abiotic stress, promoter regions (2000 bp upstream) of all 18 *LuWOX* genes were analyzed, excluding common elements like TATA-box and CAAT-box. A total of 461 cis-acting elements were predicted and categorized into four classes: light-responsive (180 elements, 39.05%), hormone-responsive (132 elements, 28.63%), stress-related (102 elements, 22.13%), and development-related (46 elements, 9.98%) (Figure 5A–C; Appendix A). Light-responsive cis-elements predominated, featuring Box 4, G-box, and TCT-motif. Hormone-regulatory elements included MeJA-responsive (TGACG/CGTCA), ABA-associated ABRE, and GA-related motifs (P-box/GARE/TATC), as well as auxin-responsive (TGA-element) and salicylic-acid-responsive (TCA-element) elements, with MeJA-related motifs being the most abundant. Stress-related elements included anaerobic induction (ARE), drought-inducible MYB-binding (MBS), CCAAT-box, TC-rich repeat, and low-temperature-responsive (LTR) elements. Development-related elements featured meristem expression (CAT-box), zein metabolism regulation (O2-site), protein-binding (AT-rich element, HD-Zip 3), and circadian rhythm (MSA-like) motifs. These findings suggest that *LuWOX* genes are intricately regulated by environmental cues, hormonal signals, and developmental programs, positioning them as key players in flax stress adaptation and growth regulation.

### 2.5. LuWOX Interactome Network and Gene Ontology Enrichment Profiling

To unravel the biological functions and regulatory networks of the *LuWOX* family, a protein–protein interaction (PPI) network was constructed using *Arabidopsis* WOX homologs as reference (Figure 6A). The analysis identified eight *LuWOX* genes interacting with seven *Arabidopsis* functional genes: AP2 (involved in dehydration and cold-induced gene expression), LFY, RPL (linked to disease resistance), TFL2, UFO, SEU, and SUP (associated with meristem differentiation and floral development). Within the *LuWOX* family, only two interaction pairs were detected: *LuWOX11*–*LuWOX5* and *LuWOX11*–*LuWOX17*. GO enrichment analysis further revealed that *LuWOX* genes are predominantly enriched in biological processes such as cell population proliferation, meristem development, anatomical structure formation during morphogenesis, leaf/phyllome development, and embryo development (Figure 6B). These results implicate *LuWOX* genes in critical developmental pathways and stress adaptation, offering mechanistic insights into their regulatory roles in flax growth and environmental responses.

### 2.6. Analysis of LuWOX Gene Expression Pattern Based on RNA-Seq Data

To delineate the expression dynamics of the *LuWOX* gene family in flax development, RNA-seq data were analyzed across diverse tissues (Figure 7). Tissue-specific expression patterns were observed, with 50% of *LuWOX* genes showing high expression in flowers at 5 days post anthesis (DPA) and 44.44% in flowers at 10 DPA (Figure 7A). Notably, *LuWOX05* and *LuWOX11* remained highly expressed in flowers at 30 DPA whereas all *LuWOX* genes exhibited low expression in floral tissues at 20 DPA. These temporally regulated expression profiles suggest the stage-specific functional diversification of *LuWOX* genes during floral development, potentially linked to their roles in late-phase reproductive processes.

To explore the potential roles of *LuWOX* genes in abiotic stress responses, their expression profiles under salt and heat stress were analyzed using publicly available transcriptomic data (Figure 7B). Heat stress induced no significant changes in most *LuWOX* genes, except for *LuWOX15*, which showed marked upregulation in stem tissues. Conversely, salt stress globally suppressed *LuWOX* expression compared to untreated controls, with tissue-specific exceptions: *LuWOX03*, *LuWOX11*, and *LuWOX17* were significantly upregulated in roots while *LuWOX02* exhibited enhanced expression in leaves. Strikingly, salt responsiveness varied across tissues: seven genes (*LuWOX01*, *LuWOX04*, *LuWOX08*, *LuWOX09*, *LuWOX11*, *LuWOX16*, *LuWOX17*, and *LuWOX18*) were downregulated in roots and five genes (*LuWOX01*, *LuWOX04*, *LuWOX07*, *LuWOX13*, and *LuWOX14*) were suppressed in leaves. These findings highlight the salt-sensitive nature of *LuWOX* genes and their tissue-specific regulatory roles, suggesting their involvement in flax stress adaptation pathways.

The expression profiling of *LuWOX* genes across 14 flax tissues using published transcriptomic data (Figure 7C) revealed tissue-specific expression patterns. *LuWOX02* was highly expressed in torpedo embryos while *LuWOX05*/*LuWOX17* dominated in mature embryos, *LuWOX04*/*LuWOX05* in heart embryos, *LuWOX02*/*LuWOX17* in cotyledon embryos, and *LuWOX10*/*LuWOX11* in anthers. Strikingly, *LuWOX01* and *LuWOX02* exhibited broad expression in vegetative and reproductive tissues (roots, stems, leaves, ovaries, pistils, stamens, fruits, and seeds) whereas *LuWOX08* and *LuWOX16* were preferentially expressed in roots and stems. The clustered expression of *LuWOX* genes during embryonic stages (torpedo, mature, heart, and cotyledon) strongly implies their conserved roles in embryo development and plant morphogenesis. These spatiotemporal expression dynamics underscore the functional versatility of *LuWOX* genes across flax developmental programs.

### 2.7. Expression Level of LuWOX Gene in Flax Under Abiotic Stress

To validate the roles of *LuWOX* genes in abiotic stress responses, qRT-PCR was performed to analyze their expression in leaf tissues under salt, cold, and drought stress at 0, 6, 12, and 24 h (Figure 8, Figure 9 and Figure 10). Under salt stress, 10 genes (*LuWOX3*, *LuWOX5*, *LuWOX6*, *LuWOX9*, *LuWOX11*, *LuWOX13*, *LuWOX14*, *LuWOX16*, *LuWOX17*, and *LuWOX18*) exhibited significant upregulation at 3 h, peaking with expression levels 3.8- to 14-fold higher than controls. Four genes (*LuWOX1*, *LuWOX2*, *LuWOX7*, and *LuWOX8*) reached peak upregulation at 12 h (14-, 11-, 6.8-, and 4.5-fold increases, respectively). Conversely, three genes (*LuWOX4*, *LuWOX10*, and *LuWOX15*) were downregulated, with minimal expression at 6–12 h, suggesting potential negative regulatory roles in salt adaptation. These dynamic expression profiles highlight the temporal specificity of *LuWOX* genes in mediating salt stress responses, offering candidate targets for improving flax salinity tolerance.

Under prolonged cold stress, the expression of *LuWOX* genes exhibited significant temporal dynamics (Figure 9). Fifteen genes (*LuWOX2*–*LuWOX14*, *LuWOX16*, and *LuWOX18*) showed rapid upregulation, peaking at 3 h with 6.3- to 35.6-fold increases compared to controls (6.3-, 8.3-, 8.1-, 24.2-, 19.6-, 16.8-, 28.8-, 19.6-, 14.4-, 35.6-, 5.7-, 9.4-, 8.1-, 6.7-, and 4.2-fold, respectively). *LuWOX1* and *LuWOX15* displayed delayed peaks at 24 h (6.4-fold) and 9 h (11.8-fold), respectively. In contrast, *LuWOX17* was downregulated, reaching minimal expression at 12 h (4.2-fold reduction), suggesting its potential role in the negative feedback regulation of cold stress adaptation.

Prolonged drought stress triggered significant temporal shifts in *LuWOX* gene expression (Figure 10). Four genes (*LuWOX3*, *LuWOX6*, *LuWOX7*, and *LuWOX8*) were rapidly upregulated, peaking at 3 h with 2.1-, 21.6-, 21.1-, and 9.3-fold increases, respectively. In contrast, 10 genes (*LuWOX1*, *LuWOX2*, *LuWOX5*, *LuWOX9*, *LuWOX11*, *LuWOX13*, *LuWOX14*, *LuWOX16*, *LuWOX17*, and *LuWOX18*) exhibited delayed upregulation, reaching maximal expression at 24 h with 32.1-, 25.4-, 6.7-, 28.6-, 25.2-, 21.1-, 24.2-, 13.8-, 22.1-, and 7.4-fold increases, respectively, compared to non-stressed controls.

## 3. Discussion

The WOX genes encode transcription factors (TFs) within the homeodomain superfamily, distinguished by a conserved helix–turn–helix DNA-binding motif. These TFs are integral to orchestrating plant developmental programs and mediating adaptive responses to biotic and abiotic environmental challenges [28,29,30]. Although there have been reports on WOX in other types, there is little information about flax WOX. So far, there has been no study using genomics to identify the *LuWOX* gene family in flax. The flax genome harbored 18 *LuWOX* genes, a number significantly smaller than those reported in *Camellia sinensis* (43) [31] and *Glycine max* (33) [21]. However, this count was comparable to the numbers in *Zea mays* (21) [32], *Paeonia suffruticosa* (22) [33], and *Helianthus annuus* (18) [34] and slightly higher than in *Sorghum bicolor* (11) [32], *Jatropha curcas* (10) [35], *Vitis vinifera* (12) [36], and *Cucumis sativus* (11) [37]. These comparative genomic insights highlight the lineage-specific expansion and contraction dynamics of WOX gene families across plant species. Phylogenetic analysis revealed that the *LuWOX* genes were classified into three clades (Ancient, Intermediate, and WUS Clades) (Figure 1), a categorization consistent with evolutionary patterns observed in WOX families of other plant species [33,34,37].

Motif composition analysis demonstrated that motif1, representing the core WOX domain, was universally conserved across all *LuWOX* proteins (Figure 2B), a pattern consistent with findings in *Melastoma dodecandrum* Lour [33]. This further demonstrates that the WOX gene family is highly conserved across plant species. The exon–intron structure analysis of *LuWOX* genes revealed that exon numbers ranged from two to four while intron numbers varied between one and three (Figure 2C). However, substantial structural divergence was observed among most genes. These structural variations in WOX genes were likely attributed to exon/intron gain/loss mechanisms during evolution [38]. Gene duplication events played pivotal roles in the evolution of WOX genes, with duplicated members potentially acquiring novel functions that enhanced plant adaptive responses to environmental stresses [39]. In this study, the 18 *LuWOX* genes were unevenly distributed across 12 chromosomes in flax, with no tandem duplication events detected (Figure 3). However, nine pairs of segmentally duplicated collinear relationships were identified (Figure 4A), strongly suggesting that segmental duplication served as the primary driver for the expansion of the *LuWOX* gene family. This evolutionary mechanism aligns with observations in other plant species, where segmental duplication facilitates functional diversification and adaptive evolution under selective pressures, particularly in stress-responsive gene families [40]. Collinearity analysis proved to be a powerful method for analyzing the evolutionary trajectories of genes [40]. Comparative collinearity analyses with *Arabidopsis*, *Zea mays*, and *Oryza sativa* identified ten, ten, and six collinear gene pairs, respectively (Figure 4B). These conserved syntenic relationships strongly suggest that the homologous gene pairs likely originated from a common ancestral gene predating species divergence. Such evolutionary conservation underscores the critical role of segmental duplication and genome-wide polyploidization events in shaping the functional diversification and adaptive expansion of the WOX family across divergent plant lineages [41]. The results demonstrated that *LuWOX* genes are highly conserved in flax, and it was proposed that the expansion of the *LuWOX* family was predominantly attributed to segmental duplication events.

Cis-regulatory elements in promoters mediate and regulate gene expression. The interaction between transcription factors and promoter-binding sites plays a pivotal role in this process [42]. The analysis of WOX gene promoter regions revealed abundant cis-acting regulatory elements, including those associated with light response (G-box), stress adaptation (LTR), and hormone signaling pathways, among which MeJA-responsive motifs (CGTCA and TGACG) and ABA-responsive elements (ABRE) were particularly prominent (Figure 5). Notably, G-box, CGTCA, TGACG, and ABRE emerged as the most enriched motifs in promoter regions. Methyl jasmonate (MeJA), a central mediator of plant responses to biotic and abiotic stresses, and abscisic acid (ABA), which plays critical roles in salinity/drought adaptation, seed dormancy regulation, and developmental plasticity during growth, are likely key hormonal signals orchestrating *LuWOX*-mediated stress adaptation. These findings highlight the evolutionary conservation of stress-responsive cis-regulatory architectures in WOX promoters and their potential roles in integrating environmental cues with developmental programs [43,44]. These results indicate that the cis-regulatory elements of *LuWOX* genes are highly conserved and play critical roles in modulating stress-related hormonal pathways and biotic/abiotic stress responses.

Protein–protein interactions are not only critical for maintaining normal protein functionality but also serve as key determinants in predicting the functional diversity of proteins [45]. In this study, eight *LuWOX* genes were found to interact with seven functional genes from *Arabidopsis*, with the most prominent interaction observed with AP2 (Figure 6A). AP2, a major class of plant-specific transcription factors, not only regulates abiotic stress responses but also participates in the fine-tuned regulation of developmental processes, highlighting its dual role in coordinating stress adaptation and growth regulation. These interactions suggest conserved regulatory networks between flax and *Arabidopsis*, where WOX-AP2 modules may serve as critical hubs for integrating environmental and developmental signals [46]. This proves the important role of WOX genes in the plant growth and development network and the stress response network.

RNA-seq data analysis indicated that the expression of *LuWOX* genes varied across tissues. Most *LuWOX* genes showed high expression in flowers at 5 days post anthesis (DPA), and *LuWOX02* was highly expressed in torpedo embryos, *LuWOX05*/*LuWOX17* in mature embryos, *LuWOX04*/*LuWOX05* in heart embryos, *LuWOX02*/*LuWOX16* in cotyledon embryos, and *LuWOX10*/*LuWOX11* in anthers (Figure 7). Previous studies demonstrated that AtrWUS in *Arabidopsis* plays a key role in stem cell formation and promotes the transition to embryogenesis [47]. The WOX genes drive embryonic pattern formation by regulating cell fate determination, orchestrating signaling responses, and establishing stem cell niches, underscoring their central role in plant developmental biology [48]. In cotton, the majority of WOX genes were actively expressed during somatic embryogenesis, highlighting their potential roles in regulating cell reprogramming and embryonic patterning [49]. In wheat (Triticum aestivum), the overexpression of TaWOX5 significantly enhanced the transformation and regeneration efficiency of immature embryos while causing no inhibitory effects on shoot differentiation or root development, thereby overcoming the genotype dependency inherent in conventional transformation methods [50]. These results demonstrate that WOX genes play crucial regulatory roles in plant growth and development, with certain members exhibiting distinct functional specializations in specific tissues. Collectively, the tissue-specific expression profiling in flax provided deeper mechanistic insights into the functional diversification of *LuWOX* genes, advancing our understanding of their roles in orchestrating developmental programs and stress adaptation. The WOX genes, constituting a class of multifunctional regulators widely conserved across plant species, play pivotal roles not only in growth and development but also in stress adaptation. Expression profiling under salt stress revealed that *LuWOX03*, *LuWOX11*, and *LuWOX17* were significantly upregulated in root tissues while *LuWOX02* exhibited enhanced expression in leaves compared to controls (Figure 7). qRT-PCR validation further confirmed that 10 *LuWOX* genes (*LuWOX3*, *LuWOX5*, *LuWOX6*, *LuWOX9*, *LuWOX11*, *LuWOX13*, *LuWOX14*, *LuWOX16*, *LuWOX17*, and *LuWOX18*) showed rapid upregulation at 3 h post salt treatment (Figure 8). Under cold stress, 15 genes (*LuWOX2*–*LuWOX14*, *LuWOX16*, and *LuWOX18*) reached peak expression at 3 h, whereas *LuWOX1* and *LuWOX15* were maximally induced at 24 h and 9 h, respectively (Figure 9). Similarly, drought stress triggered the transient upregulation of four genes (*LuWOX3* and *LuWOX6*–*LuWOX8*) at 3 h and delayed the activation of 10 genes (*LuWOX1*, *LuWOX2*, *LuWOX5*, *LuWOX9*, *LuWOX11*, *LuWOX13*, *LuWOX14*, and *LuWOX16*–*LuWOX18*) at 24 h in leaf tissues (Figure 10). These findings underscore the stress-specific temporal dynamics of *LuWOX* genes, highlighting their roles in mediating early and late responses to abiotic stresses. The distinct expression patterns suggest functional diversification among *LuWOX* members, enabling flax to fine-tune stress adaptation mechanisms across tissues and developmental stages. A previous study demonstrated that the ectopic expression of *JcWOX5* in rice enhanced drought sensitivity, suggesting a potential role of WOX genes in modulating stress tolerance pathways across divergent plant species [51]. *PagWOX11/12a* was strongly induced in roots under drought and salt stress conditions, and the overexpression of PagWOX genes in poplar significantly enhanced salt and drought tolerance [14]. *MdWOX13-1* directly transcriptionally regulated the expression of the reactive oxygen species (ROS) scavenging enzyme *MdMnSOD*, significantly enhanced the cellular ROS scavenging capacity, and improved drought stress adaptation in plants [52]. These findings provided robust evidence that WOX genes play crucial roles in abiotic stress responses in flax.

## 4. Materials and Methods

### 4.1. Plant Material

An experiment was conducted with flax variety “longya10” [53]. First we disinfected the seeds with 75% ethanol for 10 min, then washed them with sterile water and finally transplanted them into the soil. The temperature and lighting duration of the incubator were set to 26/18 °C and 16/8 h. Stress treatments were carried out when flax seedlings grew to 6–7 cm. For the drought and salt treatment groups, we extracted the plants from the soil, thoroughly rinsed them with clean water, and placed them into conical flasks containing 10% polyethylene glycol and 100 mM sodium chloride solutions, respectively. The control group was placed in clean water. Low-temperature treatment (4 °C) was applied with synchronized sampling at 0, 6, 12, and 24 h post-treatment. Triplicate biological replicates were collected concurrently across treatment groups to eliminate circadian rhythm interference. All flax specimens were flash-frozen in liquid nitrogen and cryopreserved at −80 °C.

### 4.2. Identification of WOX Gene in Flax

The whole flax genome sequence was downloaded from NCBI (entry number QMEI02000000, accessed on 3 October 2024) and the genome annotation was downloaded from Figshare (https://figshare.com/articles/dataset/Annotation_files_for_Longya-10_genome/13614311, accessed on 3 October 2024) [54]. We obtained *Arabidopsis* WOX protein sequence from *Arabidopsis* information resource library (https://www.arabidopsis.org/, accessed on 3 October 2024) [54]. Using *Arabidopsis* WOX protein sequence as query sequence, BLASTp search was performed on all protein sequences of flax (e value was set to 1 × 10^−10^) [55]. The Hidden Markov Models (PF00046) were obtained from the Pfam database (http://pfam.xfam.org/, accessed on 3 October 2024) [56]. Flax WOX genes were also predicted by hmmsearch program of HMMER3.0 software [57]. The *LuWOX* genes were filtered and corrected using CDD (https://www.ncbi.nlm.nih.gov/cdd/, accessed on 4 October 2024). A total of 18 *LuWOX* genes were systematically identified. Computational characterization of their physicochemical parameters—including coding sequence (CDS) length, amino acid residue count, molecular mass (Da), theoretical isoelectric point (pI), and Grand Average of Hydropathicity (GRAVY)—was performed using the ExPASy ProtParam platform (https://web.expasy.org/protparam/; accessed on 4 October 2024) [58]. The WoLF PSORT net station was used to carry out the subcellular location pre-test (https://wolfpsort.hgc.jp/, accessed on 4 October 2024).

### 4.3. Phylogeny, Chromosome Location, Conserved Domain, and Conserved Motif of LuWOX Gene

WOX protein sequences of *Arabidopsis* were retrieved from The *Arabidopsis* Information Resource (TAIR, https://www.arabidopsis.org/; accessed 10 October 2024) while the complete proteome of *Oryza sativa* was acquired from Phytozome v13 (https://phytozome-next.jgi.doe.gov/; accessed 10 October 2024) [58]. ClustalW in MEGA (version 11) uses default parameters to compare the WOX amino acid sequences of *Arabidopsis*, rice, and flax. Based on the maximum likelihood method (ML), a phylogenetic tree was constructed using MEGA version 11 and default configuration (neighborhood connection; parameter: Bootstrap1000) [59]. The *LuWOX* gene was located on the chromosome by flax genomic FASTA file and gff3 annotation file. The CD-search website in NCBI (https://www.ncbi.nlm.nih.gov/Structure/cdd/wrpsb.cgi, accessed on 10 October 2024) predicted its conservative structural region. Using MEME (https://meme-suite.org/meme/tools/meme, accessed on 10 October 2024), motif analysis of *LuWOX* amino acid motifs [60] and visualization using TBtools (version 2.069) were completed [61].

### 4.4. Genome-Wide Replication and Collinear Analysis of LuWOX Gene

We downloaded the genome and annotation files of *Arabidopsis* from the public-database TAIR website (https://www.arabidopsis.org/, accessed on 10 October 2024). Multiple collinear scanning (MCScanX) toolkits were used to predict collinear relationships [62]. The repetitive *LuWOX* gene was identified as genome-wide repeat (WGD). Tandem repeat genes were two or more homologous genes on chromosomes where the distance was not more than 100 kb and there were no other genes. The fragment repeat gene (Score < 1 × 10^−5^) was detected by nucleotide BLAST 2.14.0 (BLASTN), which contained a range of 100 kb (upstream and downstream 50 kb) around the coding sequence (CDS). The criteria for identifying repetitive genes included a sequence alignment length of at least 200 bp and a sequence similarity greater than 85% [63].

### 4.5. Cis-Acting Element Analysis

The genomic sequences upstream (2000 bp) of all *LuWOX* genes were extracted using TBtools software (version 2.069). Cis-regulatory elements were predicted via the PlantCARE online platform (http://bioinformatics.psb.ugent.be/webtools/plantcare/html/, accessed on 15 October 2024) [64], and the results were visualized using TBtools (version 2.069).

### 4.6. Construction of Protein Interaction Network and GO Enrichment Analysis

Protein–protein interaction (PPI) network analysis of flax WOX proteins and *Arabidopsis* stress-responsive genes was performed using the STRING database (https://string-db.org/, accessed on 21 October 2024). The interaction networks were visualized with Cytoscape software (v3.9.1). For GO enrichment analysis of *LuWOX* genes, the GO-base.ob file was retrieved via TBtools (version 2.069), and eggNOG-mapper (http://eggnog-mapper.embl.de/, accessed on 21 October 2024) [65] was employed to annotate flax genomic protein sequences. Final visualization of GO terms was conducted using TBtools (version 2.069).

### 4.7. Analysis of Expression Pattern of LuWOX Gene Family

This study sequenced the transcriptomes of five flax species: (I) pistil, stamen, fruit, and stem tip tissues (PRJNA1002756) (https://www.ncbi.nlm.nih.gov/sra/?term=, accessed on 25 October 2024); (ii) flower tissue at 30, 20, 10, and 5 days after flowering (PRJNA833557); (iii) different flax embryo tissues, anthers, and seed tissues (PRJNA663265); (iv) root and leaf tissues after salt stress (PRJNA977728); and (v) stem tissue after heat stress (PRJPA874329). After filtering, the data were aligned with the Longya10 reference genome, and expression quantification was performed using tidyverse [66], Rsubrad [67], limma [68], and edgeR [69] R packages. Finally, a heatmap of log_2_(FPKM) values was generated using TBtools (version 2.069).

### 4.8. RNA Extraction and Fluorescence Quantitative PCR Analysis

Total RNA isolation was performed using the SPARKeasy (AC0201) RNA extraction kit (China Shandong Sparkjade Biotechnology Co., Ltd., Shandong, China), followed by reverse transcription into complementary DNA (cDNA). Quantitative reverse-transcription PCR (qRT-PCR) assays were conducted with the TB Green™ Premix Ex Taq™ II system (TaKaRa Bio, Kyoto, Japan), employing 20 μL reaction volumes containing gene-specific primers (Appendix A) designed via Primer Premier 5.0 software. Triplicate technical replicates were analyzed per biological sample, normalized against the endogenous GAPDH (glyceraldehyde-3-phosphate dehydrogenase) reference gene. Relative expression quantification was performed using the 2^−ΔΔCt^ method.

## 5. Conclusions

This paper presents the first comprehensive analysis of the WOX gene family in flax. A total of 18 *LuWOX* genes were identified in the flax genome and phylogenetically classified into three clades (Ancient, Intermediate, and WUS Clades). Conserved exon–intron structures and motif compositions were observed among members within the same clade, suggesting functional conservation. Collinearity analysis revealed eight segmentally duplicated *LuWOX* gene pairs, indicating that segmental duplication drove family expansion. Protein–protein interaction networks implicated several *LuWOX* proteins in abiotic stress-responsive pathways. Promoter analysis demonstrated that most *LuWOX* genes harbor hormone- (MeJA, ABA) and abiotic stress-related cis-regulatory elements. Expression profiling and qRT-PCR validation revealed that *LuWOX* genes are dynamically expressed during developmental processes (5-day-post-anthesis flowers and embryos) and in response to hormonal cues and abiotic stresses (cold, drought, and salinity). These findings systematically link *LuWOX* gene structure, evolution, and stress responsiveness, offering actionable targets for molecular breeding. Beyond flax, the conserved roles of WOX genes in balancing development and stress adaptation could inform resilience breeding in climate-vulnerable crops, contributing to sustainable agriculture under global change.

## Figures and Tables

**Figure 1 ijms-26-03571-f001:**
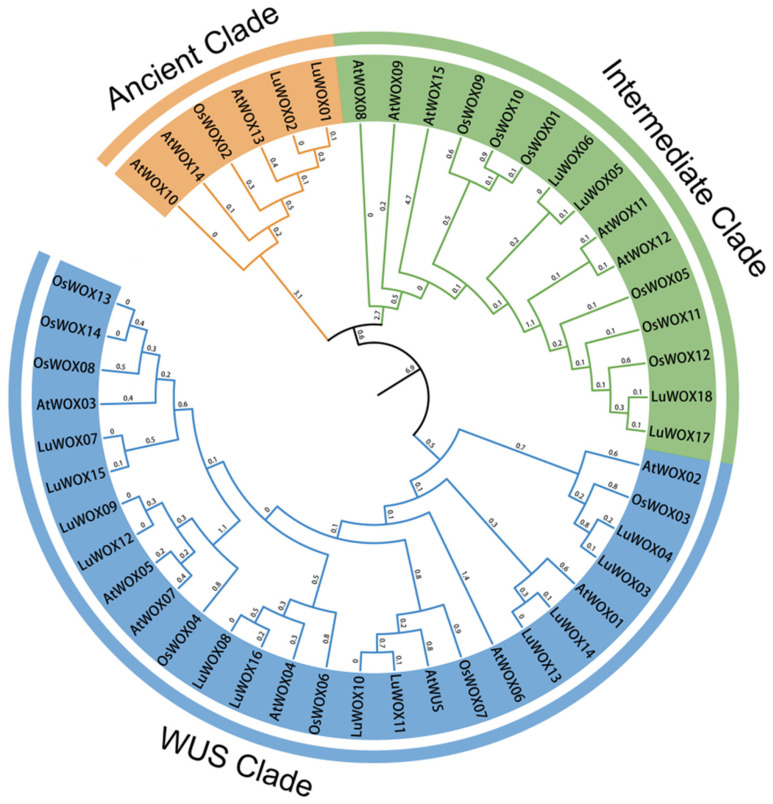
WOX protein phylogenetic tree of three species. The At prefix represents *Arabidopsis*, the Os prefix represents rice, and the Lu prefix represents flax. The WOX gene can be divided into three subfamilies, represented by different colors.

**Figure 2 ijms-26-03571-f002:**
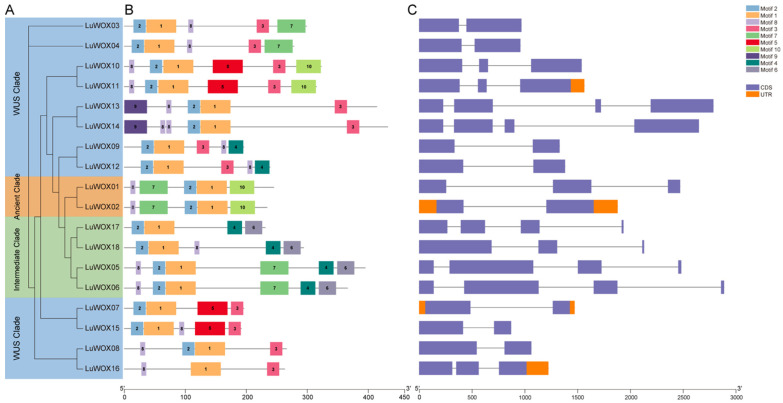
Structural and evolutionary features of *LuWOX* genes. (**A**) Maximum likelihood phylogeny of *LuWOX* proteins. (**B**) Conserved motif architecture with 10 identified motifs color-coded. Genomic sequence spans are denoted by gray connectors. (**C**) Exon–intron organization with coding sequences (CDS, purple), untranslated regions (UTR, orange), and intronic spacers (black).

**Figure 3 ijms-26-03571-f003:**
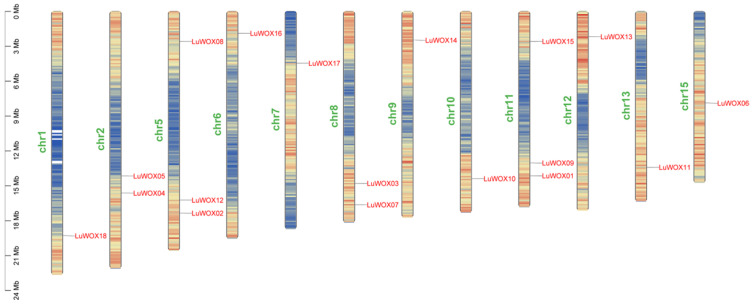
Chromosomal localization of *LuWOX* genes. Genomic distribution visualized using a 100 kb sliding window, with gene density depicted by a gradient color scale from red (high) to blue (low).

**Figure 4 ijms-26-03571-f004:**
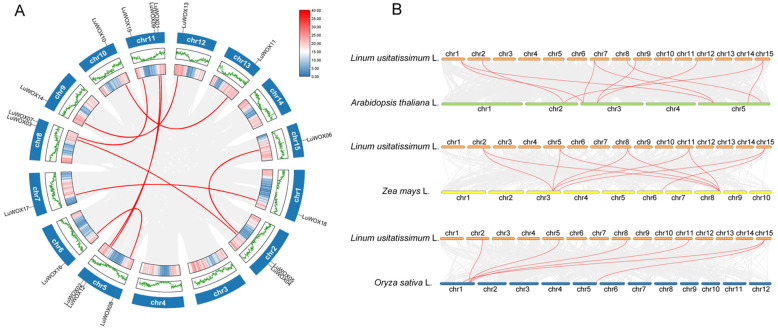
Syntenic relationships of *LuWOX* genes. (**A**) Intraspecific synteny of *LuWOX* genes in flax. (**B**) Interspecific synteny analysis between flax and *Arabidopsis*, maize, rice. Red connectors designate *LuWOX* orthologous pairs, gray linkages represent genome-wide syntenic blocks.

**Figure 5 ijms-26-03571-f005:**
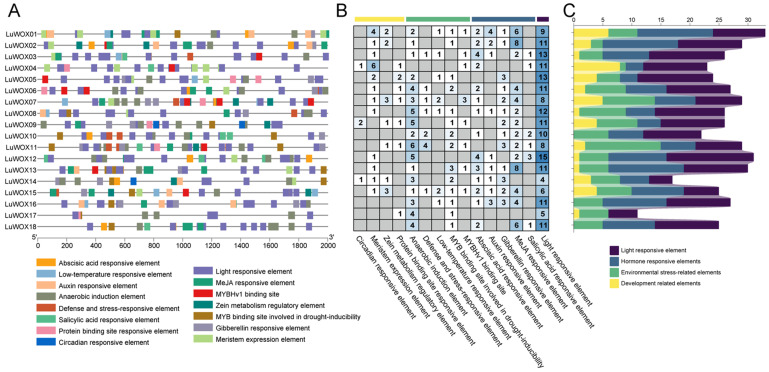
Cis-regulatory element profiling of *LuWOX* genes. (**A**) Element distribution across three *LuWOX* subfamilies. (**B**,**C**) Element quantification categorized by function: yellow (developmental), green (stress-responsive), dark blue (hormonal), and purple (light-related).

**Figure 6 ijms-26-03571-f006:**
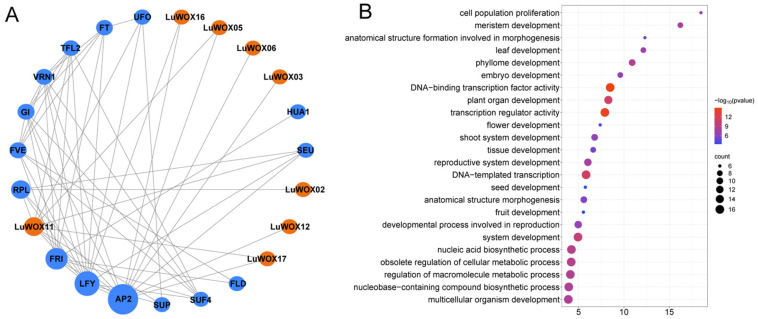
*LuWOX* interactome network and GO enrichment profilin. (**A**) *LuWOX* family interactome mapped through *Arabidopsis* orthologs. Blue represents resistance related genes in *Arabidopsis* while orange represents the *LuWOX* gene family. (**B**) GO enrichment analysis of *LuWOX* gene family.

**Figure 7 ijms-26-03571-f007:**
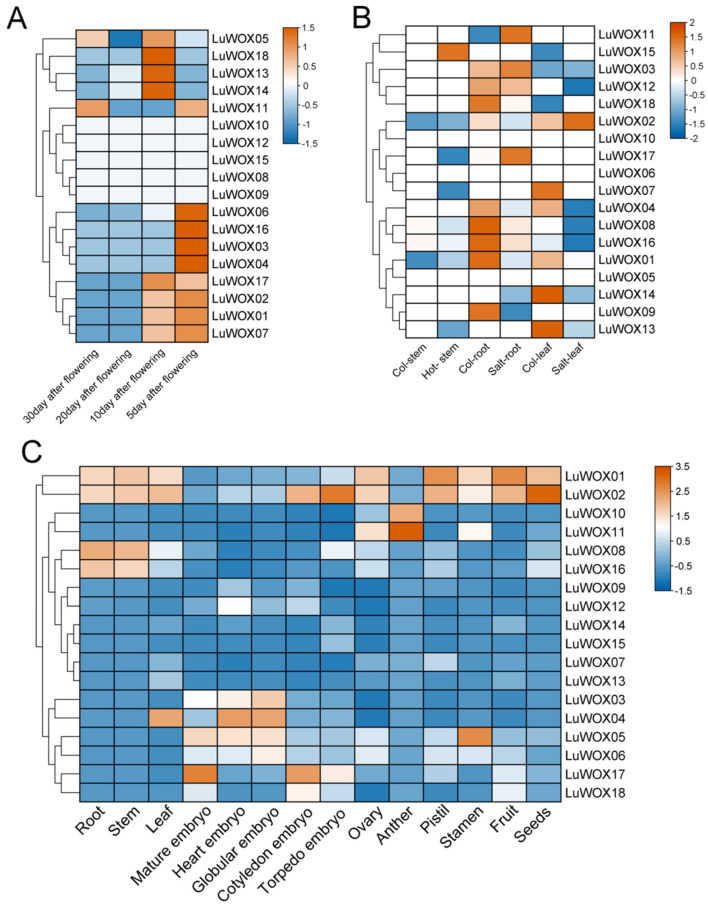
Transcriptional profiling of *LuWOX* gene family. (**A**) Post-anthesis floral expression dynamics. (**B**) Abiotic stress responses (salt/heat). Col-Stem, Col-root, and Col-leaf represent the stem, root, and leaf tissues of the experimental control group, respectively. (**C**) Tissue-specific expression atlas in flax. Expression levels are log_2_(FPKM+1)-transformed, visualized via gradient color scaling from high (orange) to low (blue).

**Figure 8 ijms-26-03571-f008:**
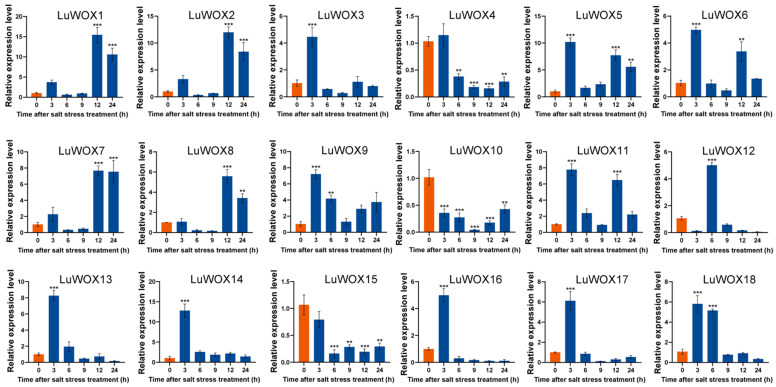
Salt-stress-induced *LuWOX* transcriptional dynamics: untreated controls (blue) versus NaCl-treated samples (orange). Statistical significance is denoted by asterisks (Student’s *t*-test) (** *p* < 0.01; *** *p* < 0.001).

**Figure 9 ijms-26-03571-f009:**
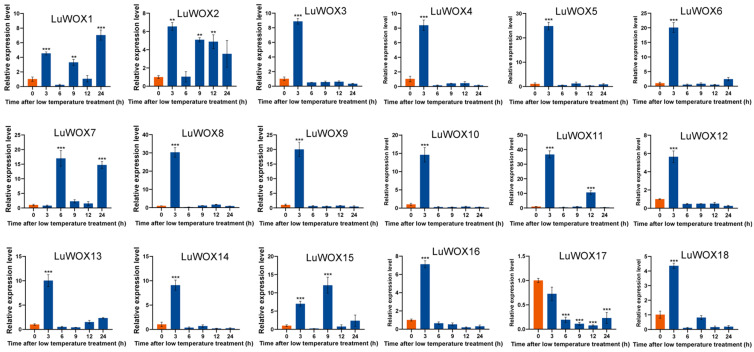
Cold-stress-induced *LuWOX* transcriptional dynamics: untreated controls (blue) versus NaCl-treated samples (orange). Statistical significance denoted by asterisks (Student’s *t*-test) (** *p* < 0.01; *** *p* < 0.001).

**Figure 10 ijms-26-03571-f010:**
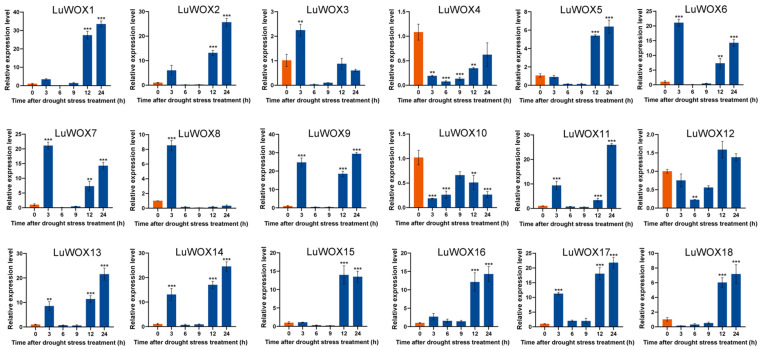
Drought-stress-induced *LuWOX* transcriptional dynamics: untreated controls (blue) versus NaCl-treated samples (orange). Statistical significance is denoted by asterisks (Student’s *t*-test) (** *p* < 0.01; *** *p* < 0.001).

**Table 1 ijms-26-03571-t001:** Prediction and characterization of WOX genes in flax.

Gene	Gene ID in Genome	Number of Amino Acids	Molecular Weight (KDa)	PI	Instability Index	Aliphatic Index	Grand Average of Hydropathicity (GRAVY)	Subcellular Localization
LuWOX01	L.us.o.m.scaffold170.48	245	26.88	6.75	39.55	70.08	−0.629	Nuclear
LuWOX02	L.us.o.m.scaffold255.15	234	25.94	5.63	55.21	65.04	−0.832	Nuclear
LuWOX03	L.us.o.m.scaffold218.7	299	33.44	8.69	54.07	63.95	−0.638	Nuclear
LuWOX04	L.us.o.m.scaffold50.47	278	30.72	8.76	63.10	60.72	−0.764	Nuclear
LuWOX05	L.us.o.m.scaffold33.69	395	43.91	7.38	56.58	59.65	−0.677	Nuclear
LuWOX06	L.us.o.m.scaffold107.61	366	40.69	7.86	60.13	59.59	−0.763	Nuclear
LuWOX07	L.us.o.m.scaffold155.99	196	22.01	9.64	74.39	63.72	−0.793	Nuclear
LuWOX08	L.us.o.m.scaffold98.176	266	29.85	8.69	56.80	58.68	−0.973	Nuclear
LuWOX09	L.us.o.m.scaffold8.54	196	22.58	8.42	51.73	56.28	−0.825	Nuclear
LuWOX10	L.us.o.m.scaffold133.89	323	35.57	5.68	50.37	47.43	−0.930	Nuclear
LuWOX11	L.us.o.m.scaffold83.172	315	34.71	5.56	51.80	47.40	−0.909	Nuclear
LuWOX12	L.us.o.m.scaffold2.503	239	27.73	8.69	63.42	51.84	−0.954	Nuclear
LuWOX13	L.us.o.m.scaffold72.52	414	45.71	7.10	59.57	61.55	−0.698	Nuclear
LuWOX14	L.us.o.m.scaffold103.108	432	47.91	7.44	52.36	62.13	−0.756	Nuclear
LuWOX15	L.us.o.m.scaffold258.26	192	21.81	9.71	54.33	70.10	−0.741	Nuclear
LuWOX16	L.us.o.m.scaffold212.30	263	29.86	9.37	64.60	63.80	−0.983	Nuclear
LuWOX17	L.us.o.m.scaffold6.173	231	24.66	5.65	64.59	67.97	−0.345	Nuclear
LuWOX18	L.us.o.m.scaffold141.67	294	31.13	5.96	61.38	66.73	−0.366	Nuclear

**Table 2 ijms-26-03571-t002:** Ka, Ks, and Ka/Ks values of replication pairs of WOX gene family in flax.

Duplicated Gene Pairs	Non-Synonymous (Ka)	Synonymous (Ks)	Ka/Ks
*LuWOX01* & *LuWOX02*	0.07464	0.3158	0.2363
*LuWOX04* & *LuWOX03*	0.08564	0.2392	0.3579
*LuWOX06* & *LuWOX05*	0.04869	0.2712	0.1795
*LuWOX15* & *LuWOX07*	0.08562	0.1661	0.5151
*LuWOX08* & *LuWOX16*	0.04944	0.0838	0.5899
*LuWOX09* & *LuWOX12*	0.04245	0.2428	0.1748
*LuWOX10* & *LuWOX11*	0.02632	0.0976	0.2697
*LuWOX13* & *LuWOX14*	0.06725	0.1652	0.4069
*LuWOX17* & *LuWOX18*	0.02931	0.1340	0.2186

## Data Availability

Data is contained within the article and Appendix A.

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
