# Peer review of "Identification and Characterization of WOX Gene Family in Flax (Linum usitatissimum L.) and Its Role Under Abiotic Stress"

_ijms, 2025, doi:10.3390/ijms26083571_

Round 1

Reviewer 1 Report

Comments and Suggestions for Authors

The article entitled " Identification and Characterization of The WOX Gene Family 2 in Flax (Linum usitatissimum L.) and Its Role Under Abiotic Stress " has identified the WOX genes involed abiotic stresses in Flax. Some issues need to be concerned.

1. The authors may do more experiment validations, such as subcellular localization, In vivo stress tolerance assay in yeast, RNAi, transgenic Arabidopsis, beside the qRT-PCR. 
2. How about the Ka/Ks values among these WOX genes?
3. Table 2, For the miRNA identified for the targets of WOX genes, many of the  expectation value > 3 , which indicate the WOX genes as miRNA targets are unreliable. There are many mismatch between WOX genes and miRNAs.
4. Some figures were not clear, which should be high resolution. Figure 4B, Figure 6A, Figure 10 should be clear and suitable size for the word size. 
5. In Figure 7B, what do the "Col-stem, Col-root, Col-leaf " represent?
6. The English language should be polished because of some errors. For instance, 
(1) Table 2. "MiRNA" to "miRNA" "LuWOX gene" to "LuWOX genes".
(2) line 321," (WUSCHEL-related homeobox) " may delete. 
(3) "transcription factors (TFs)" abbreviations should appear for the first time . 

Comments on the Quality of English Language

The English language should be polished.

Author Response

Comments 1: The authors may do more experiment validations, such as subcellular localization, In vivo stress tolerance assay in yeast, RNAi, transgenic Arabidopsis, beside the qRT-PCR.

Response 1: Thank you for your thorough review and valuable suggestions on our manuscript. The proposed additional experimental validations (e.g., subcellular localization, in vivo yeast stress tolerance assays, RNAi, and transgenic Arabidopsis) are of significant scientific importance, and we fully agree that these experiments could provide more comprehensive support for the research conclusions. However, due to current time constraints in the research, we are unable to conduct all the suggested experiments at this stage. It should be noted that the core objective of this paper is to identify the flax WOX family. The existing data (qRT-PCR validation) can preliminarily support the main conclusions. Regarding the in-depth mechanistic questions raised, we plan to further explore them in subsequent studies.

Comments 2: How about the Ka/Ks values among these WOX genes?

Response 2: Thank you for reviewing our article and for your valuable feedback.

We have calculated the Ka and Ks values for the LuWOX gene pairs and added them to Table 2. Thank you once again for your review.

Comments 3: Table 2, For the miRNA identified for the targets of WOX genes, many of the  expectation value > 3 , which indicate the WOX genes as miRNA targets are unreliable. There are many mismatch between WOX genes and miRNAs.

Response 3: Thank you for your critical evaluation and constructive feedback regarding the miRNA target prediction analysis in our manuscript. We sincerely appreciate your expertise in identifying the limitations of the initial approach. As you rightly pointed out, the miRNA-WOX interactions presented in Table 2 exhibited high expectation values (E > 3) and significant sequence mismatches, which indeed compromise the reliability of these predictions. To ensure the scientific rigor of this study, we have completely removed all miRNA-related analyses from the manuscript, including Table 2 and any associated text in the Results and Discussion sections. This revision aligns with the core objective of our study, which focuses on the systematic identification and characterization of the WOX gene family in flax. The removal of miRNA predictions does not affect the validity of the primary conclusions supported by qRT-PCR and phylogenetic analyses.

Comments 4: Some figures were not clear, which should be high resolution. Figure 4B, Figure 6A, Figure 10 should be clear and suitable size for the word size.

Response 4: Thank you for highlighting the image clarity issues. We have replaced Figures 4B, 6A, and 10 with high-resolution versions and adjusted their sizes to ensure legibility of text and details. The revised figures are now embedded in the manuscript.

Comments 5: In Figure 7B, what do the "Col-stem, Col-root, Col-leaf " represent?

Response 5: We thank the reviewer for this observation and suggestion. Col-Stem, Col-root, and Col-leaf represent the stem, root, and leaf tissues of the experimental control group, respectively. We have added this information in the figure legend of Figure 7.

Comments 6: The English language should be polished because of some errors. For instance,

(1) Table 2. "MiRNA" to "miRNA" "LuWOX gene" to "LuWOX genes".

(2) line 321," (WUSCHEL-related homeobox) " may delete.

(3) "transcription factors (TFs)" abbreviations should appear for the first time .

Response 6: Thank you for your thorough review and valuable suggestions. We have carefully addressed allthe points raised regarding language polishing and technical terminology:

Reviewer 2 Report

Comments and Suggestions for Authors

Dear Authors,

The paper “Identification and Characterization of The WOX Gene Family in Flax (Linum usitatissimum L.) and Its Role Under Abiotic Stress” delivers an insightful contribution to various life sciences fields. This is the first paper to investigate the WOX Gene Family in flax, which exhibits excellent benefits for human health.

There is some inconsistency in the formatting and usage of Arabidopsis thaliana throughout the manuscript (for example – italics, full scientific name).

The role of the WOX Gene Family in plant response to abiotic stress is extensively discussed, however, its impact on biotic stress response is poorly mentioned.

In the conclusion, it would be beneficial to discuss further perspectives to highlight the study's impact, for instance, future research directions. The interpretation of the results should be expanded to involve their broader implications.

Author Response

Comments 1: There is some inconsistency in the formatting and usage of Arabidopsis thaliana throughout the manuscript (for example – italics, full scientific name).

Response 1: Thank you for highlighting the inconsistencies in the formatting of Arabidopsis thaliana. We have thoroughly revised the manuscript to ensure uniform usage. All instances have been carefully checked and corrected to align with taxonomic conventions. We appreciate your meticulous attention to detail and welcome any additional feedback.

Comments 2: The role of the WOX Gene Family in plant response to abiotic stress is extensively discussed, however, its impact on biotic stress response is poorly mentioned.

Response 2: Thank you for reviewing our article and for your valuable feedback. However, specific functional studies on WOX genes in biotic stress responses remain relatively scarce. This gap may arise because research on the WOX gene family has predominantly focused on their roles in plant development and abiotic stress adaptation. We have incorporated the findings on WOX11-related biotic stress mechanisms into the introduction section of the manuscript.

Comments 3: In the conclusion, it would be beneficial to discuss further perspectives to highlight the study's impact, for instance, future research directions. The interpretation of the results should be expanded to involve their broader implications.

Response 3: We sincerely appreciate the reviewer’s constructive feedback. As suggested, we have expanded the conclusion section to explicitly address future research directions and broader implications of our findings.

Round 2

Reviewer 1 Report

Comments and Suggestions for Authors

The authors have addressed the concerns.